# Topoarchitected polymer networks expand the space of material properties

Xiao Liu[1], Jingping Wu[1], Keke Qiao[1], Guohan Liu[1], Zhengjin Wang [1], Tongqing Lu [1], Zhigang Suo [2✉] &
Jian Hu [1✉]

Many living tissues achieve functions through architected constituents with strong adhesion.
An Achilles tendon, for example, transmits force, elastically and repeatedly, from a muscle to
a bone through staggered alignment of stiff collagen fibrils in a soft proteoglycan matrix. The
collagen fibrils align orderly and adhere to the proteoglycan strongly. However, synthesizing
architected materials with strong adhesion has been challenging. Here we fabricate archi-
tected polymer networks by sequential polymerization and photolithography, and attain
adherent interface by topological entanglement. We fabricate tendon-inspired hydrogels by
embedding hard blocks in topological entanglement with a soft matrix. The staggered
architecture and strong adhesion enable high elastic limit strain and high toughness simul-
taneously. This combination of attributes is commonly desired in applications, but rarely
achieved in synthetic materials. We further demonstrate architected polymer networks of
various geometric patterns and material combinations to show the potential for expanding
the space of material properties.

[1] State Key Lab for Strength and Vibration of Mechanical Structures, International Center for Applied Mechanics, Department of Engineering Mechanics, Xi'an
Jiaotong University, Xi'an, China. [2] John A. Paulson School of Engineering and Applied Sciences, Kavli Institute for Bionano Science and Technology, Harvard
University, Cambridge, MA, USA. ✉email: suo@seas.harvard.edu; hujian@mail.xjtu.edu.cn

Soft polymer materials, such as elastomers and gels, are under intense development to enable emerging fields of biointegration and bioinspiration, including tissue engineering[1–3], bioelectronics[4,5], and soft robots[6–8]. Many applications require soft materials to deform reversibly (high elasticity) and resist fracture (high toughness). High elasticity and high toughness, however, are often conflicting requirements in materials development. A highly elastic material loads and unloads without dissipating much energy, whereas a highly tough material resists the growth of a crack by dissipating a large amount of energy. Polymer networks have been synthesized to achieve either high elasticity or high toughness[9,10], but rarely both. The difficulty in simultaneously achieving elasticity and toughness is evident on the plane of elastic limit strain ($\varepsilon_e$, the maximum strain at which the load and unload curves coincide) and toughness ($\Gamma$, the energy consumed per unit area during crack propagation) (Fig. 1a). The two properties are negatively correlated. Highly elastic materials have low toughness ($\Gamma < 100$ J/m$^2$), and highly tough materials have low elastic limit strain

($\varepsilon_e < 100\%$)[11–23]. A large area in the top right of the plane is empty. This negative correlation originates from the commonly used toughening strategy: sacrificial bonds[10,14,24,25]. When a crack advances in such a material, the polymer network transmits high stress from the crack front to the bulk of the material, breaking sacrificial bonds in the bulk, which toughens the material. The sacrificial bonds, however, lowers the elastic limit strain.

To develop an elastic and tough material, we draw inspiration from the Achilles tendon, a tissue that transmits force elastically and repeatedly from a muscle to a bone[26]. An Achilles tendon has many parallel fascicles, and each fascicle consists of staggered collagen fibrils in a proteoglycan matrix[27] (Fig. 1b). The fibrils are stiff and the matrix is soft. They are elastic and adherent. The staggered architecture of constituents of the large difference in stiffness and strong adhesion makes the Achilles tendon both elastic and tough. Although the Achilles tendon has a modest rupture strain of ~15%, the proteoglycan matrix is subjected to considerable deformation during the extension process[28].

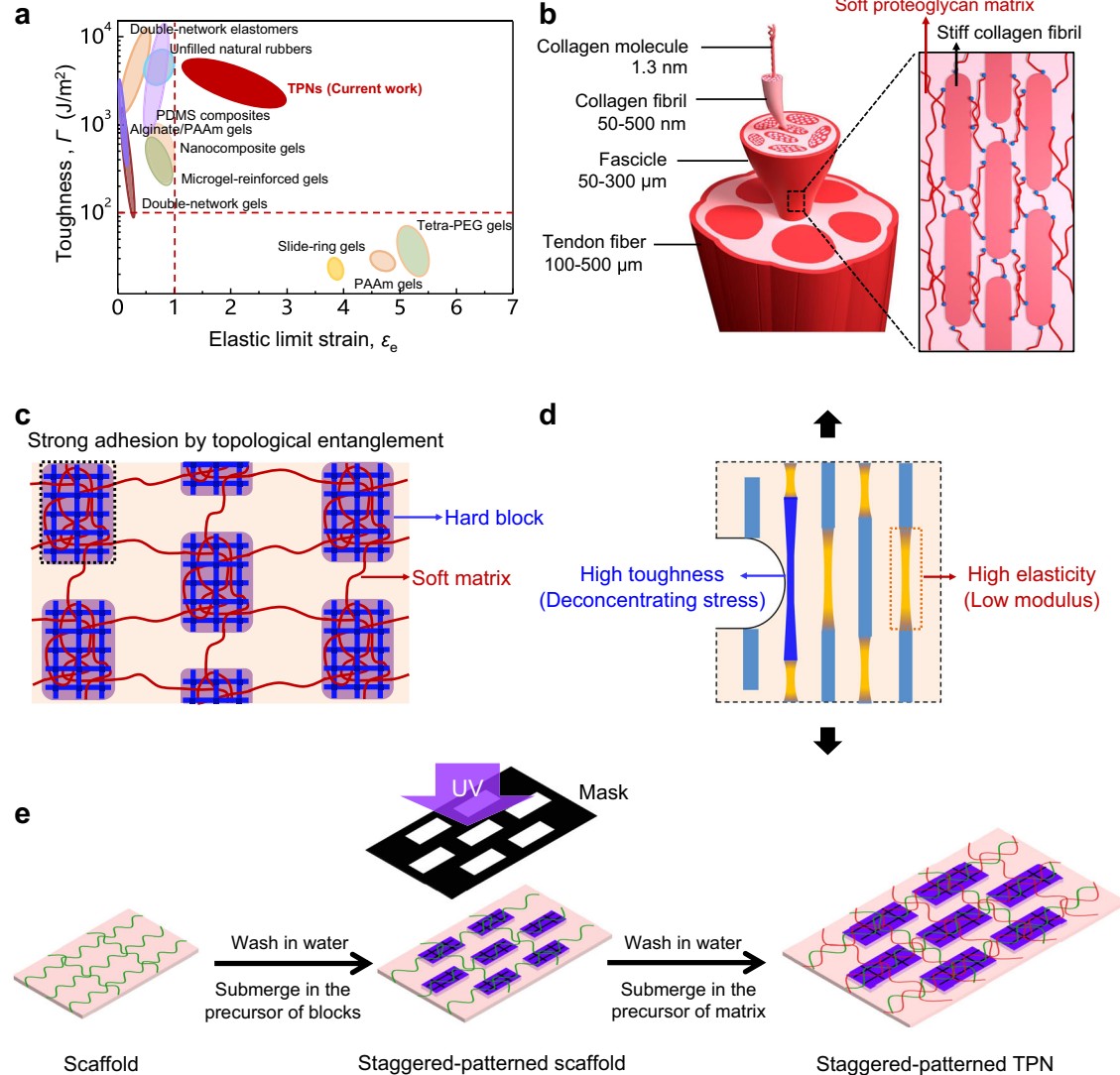

**Fig. 1 Topoarchitected polymer networks (TPNs). a** Existing elastomers and hydrogels (equilibrium-swollen in water) exhibit a negative correlation between two properties: elastic limit strain and toughness. The TPNs synthesized in this work simultaneously achieve high elastic limit strain and high toughness. **b** In a fascicle of an Achilles tendon, staggered collagen fibrils are embedded in a proteoglycan matrix. **c** Schematic illustration of TPNs. Hard blocks adhere to a soft matrix through topological entanglement of polymer networks. **d** Mechanical principle of a staggered-patterned TPN for high elasticity and high toughness. **e** TPN hydrogels prepared by sequential polymerization and photolithography.

Learning from the structure–property relationship of the Achilles tendon, here we demonstrate that a similarly architected material can achieve both high elastic limit strain and high toughness. We fabricate a staggered architecture of blocks in a matrix (Fig. 1c). Each block has a polymer network of dense crosslinks, and the matrix has a polymer network of sparse crosslinks. The polymer networks of the block and the matrix adhere strongly by topological entanglement. We call this network morphology topoarchitected polymer networks (TPNs). Different from the traditional interpenetrating polymer networks (IPNs) that are homogeneous at the scale larger than polymer meshes[29,30], TPNs are heterogeneous with architected polymer networks. When a staggered-patterned TPN is stretched, high elastic limit strain mainly comes from the soft matrix (Fig. 1d). The TPN amplifies toughness by the adherent architecture of dissimilar polymers. In a homogeneous material, a crack tip concentrates stress, which embrittles the material. By contrast, at a crack tip in a TPN, the large strain in the soft matrix spreads high stress over the length of a hard block. When the hard block breaks, the elastic energy stored in the block is released. This deconcentration of stress toughens the TPN. Hereinafter, we demonstrate the TPN principle for the elasticity-toughness integration of architected hydrogels, and the extensibility of TPNs in geometric patterns, material combinations, and multilayers.

## Results and discussion

**Preparation of TPN hydrogels**. We fabricate a class of TPN hydrogels by three-step sequential polymerization and photolithography (Fig. 1e). After each polymer network cures, the sample is submerged in water to leach unreacted ingredients. During leaching, the polymer network just synthesized swells, which must be considered in designing the fabrication process. We begin by preparing a first network of sparse crosslinks, which serves as a scaffold for the synthesis of blocks. After leaching, the scaffold is submerged in the precursor for a second polymer network, which is cured by photolithography through a mask. After leaching, the second network swells. The scaffold hosts the blocks of the second network, but is too weak to function as a matrix. We synthesize a third polymer network by submerging the staggered-patterned scaffold in another precursor, curing, and leaching. Due to the constraint of the blocks, the third polymer network swells modestly. Unless otherwise stated, the first and the third networks are sparsely crosslinked poly-acrylamide (PAAm), and the second network is highly crosslinked poly(2-acrylamido-2-methylpropanesulfonic acid) (PAMPS). In this case, the soft matrix (soft phase) consists of two interpenetrating networks (PAAm/PAAm), and hard blocks (hard phase) have three interpenetrating networks (PAAm/PAMPS/PAAm). The hydrogels at the three stages of preparation have markedly different stress–strain curves (Supplementary Fig. 1d). The single-network scaffold is soft and stretchable. The stripe-patterned scaffold is brittle. After the third network forms, the stripe-patterned TPN gel is both strong and stretchable.

We use a pattern of alternating unmasked and masked stripes to explore the effect of photolithography on the structural and mechanical properties of TPN hydrogels (Supplementary Fig. 1). The irradiation time and the stripe width in the mask affect the network morphology. The mask with the stripe width of 500 μm shows a processing window of irradiation time from 4 s to 10 s, but the one with the stripe width of 100 μm is difficult to reproduce the pattern in the gels (Supplementary Fig. 1a). After a TPN is synthesized, we immerse it in an aqueous solution of blue dye molecules, which are selectively absorbed by the second polymer network. When the irradiation time locates in the processing window, we can observe well-defined stripes,

highlighted by the blue dyes. When the irradiation time exceeds 10 s, a continuous network forms, with crosslink density in the masked region lower than that in the unmasked region, as shown by the color contrast (Supplementary Fig. 1b). The modulus, therefore, increases abruptly as the irradiation time changes from 8 s to 12 s (Supplementary Fig. 1c). For a TPN to achieve good mechanical properties, one should tune the feature size of the mask and the irradiation time.

**Deformation of TPNs**. We fabricate a TPN, stretch it using a tensile tester, and observe it through the crossed polarizer and analyzer (Fig. 2a and Supplementary Fig. 2). The TPN is composed of soft columns and composite columns in parallel. Each composite column is composed of soft and hard segments in series. The soft columns and composite columns both have the applied strain $\varepsilon$. A finite element simulation shows that a soft column undergoes both tension and shear, but a composite column undergoes tension (Supplementary Fig. 3). As the applied strain $\varepsilon$ increases, in a composite column, the soft and hard segments elongate by different strains, $\varepsilon_s$ and $\varepsilon_h$ (Fig. 2b). When the applied strain is small, $0 < \varepsilon < 1.75$, $\varepsilon_s$ increases steeply to 6, but $\varepsilon_h$ changes slowly. When the applied strain is large, $1.75 < \varepsilon < 6$, $\varepsilon_s$ changes slowly, but $\varepsilon_h$ increases to 2.4. Write $\varepsilon = \phi_s\varepsilon_s + \phi_h\varepsilon_h$, where $\phi_s$ and $\phi_h$ are the volume fraction of the soft and hard segments in the composite column. We plot the fractions of the contributions of the soft and hard segments to the applied strain, $f_s = \phi_s\varepsilon_s/\varepsilon$ and $f_h = \phi_h\varepsilon_h/\varepsilon$ (Fig. 2c). The soft segment contributes up to ~70% of the applied strain when $\varepsilon < 1.75$, but only contributes 28% of the applied strain when $\varepsilon = 6$.

We measure load and unload stress–strain curves with the increasing maximum strain (Fig. 2d). When the TPN is loaded to a maximum strain of $\varepsilon_{max} = 1.5$ and then unloaded, the load and unload stress–strain curves coincide, indicating nearly perfect elasticity. When $\varepsilon_{max} = 1.75$, the load and unload stress–strain curves differ slightly, giving a small hysteresis loop. Also plotted are the stress–strain curves of the soft gel and hard gel, which have the same chemical composition as the soft and hard phases in the TPN.

The area under the load stress–strain curve gives the work of extension, $W$, the area of the hysteresis loop gives the dissipated work, $W_D$, and the area under the unload curve gives the elastic work, $W_E$ (Fig. 2e). Note that $W = W_D + W_E$. The ratio $W_D/W$ is a dimensionless measure of hysteresis. The hysteresis is negligible when $\varepsilon_{max}$ is small, and is appreciable when $\varepsilon_{max}$ is large. As a convention, we define the elastic limit strain $\varepsilon_e$ when the hysteresis $W_D/W$ is about to exceed 1%. Our measurements give the elastic limit strain of 5 for the soft gel, 0.25 for the hard gel, and 1.5 for the TPN gel. The high $\varepsilon_e$ of the soft gel results from the entropic elasticity of long polymer chains and low viscosity of water. The low $\varepsilon_e$ of the hard gel results from the damage of the short-chain network. The TPN gel is a composite of the soft and hard gels, and has an elastic limit strain lying between its two constituent gels. The high $\varepsilon_e$ of the TPN gel is contributed dominantly by the soft segment, accounting for 68% (Fig. 2c). Like modulus and toughness, elastic limit strain is a material property of great significance to many applications. The TPNs represent a broad class of gels that enable trade-off among various material properties.

**Fracture of TPNs**. We next study the toughness of the three gels. We introduce a crack in each gel using a cutter, and stretch the gel to the critical strain $\varepsilon_c$ for rupture. We videotape the deformation process through the crossed polarizer and analyzer. For the soft gel, the crack blunts greatly, and the critical strain is large, $\varepsilon_c = 5.8$ (Fig. 3a and Supplementary Movie 1). For the hard gel,

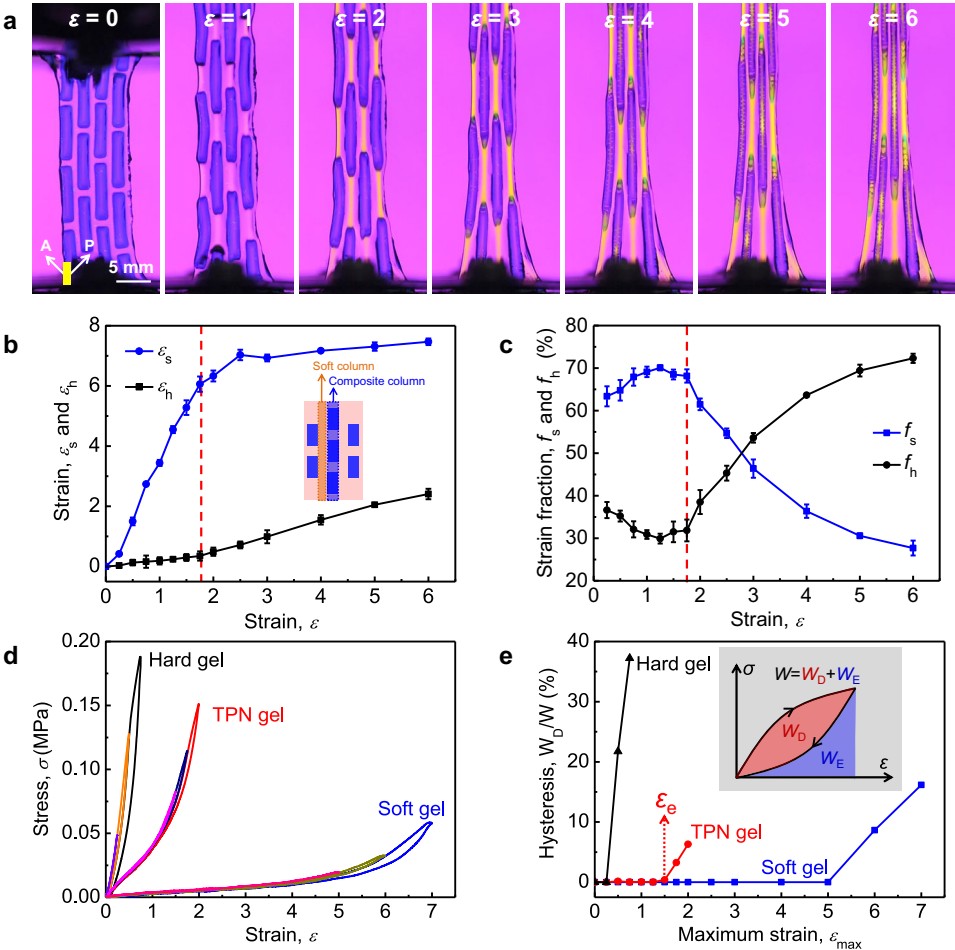

**Fig. 2 Deformation of TPNs. a** Deformation of a TPN observed using the crossed polarizer and analyzer. As the long-chain matrix deforms, the orientation-induced anisotropy of the refractive index causes the observed brightness. **b** The strain $\varepsilon_s$ and $\varepsilon_h$ of the soft and hard segments in a composite column plotted as functions of the applied strain $\varepsilon$. **c** Strain fraction $f_s$ and $f_h$ of the soft and hard segments. At least three samples were tested for the calculation of average value and standard deviation. **d** Load and unload stress–strain curves of soft, hard, and TPN gels with the increasing maximum strain. **e** Dependence of hysteresis on the maximum strain. The elastic limit strain $\varepsilon_e$ for the TPN gel is highlighted by a red dotted arrow. The inset indicates work of extension $W$, dissipated work $W_D$, and elastic work $W_E$. The ratio $W_D/W$ is a dimensionless measure of hysteresis.

the crack blunts somewhat, and the critical strain is relatively small, $\varepsilon_c = 1$ (Fig. 3b and Supplementary Movie 2). For the TPN gel, the crack also blunts significantly, and the critical strain is also large, $\varepsilon_c = 3.6$ (Fig. 3c and Supplementary Movie 3). In soft and hard gels, the birefringence contrast is insufficient to observe. In the TPN gel, as the applied strain increases, highly stretched regions light up due to birefringence. According to the birefringence distribution at the crack front, we find the soft phase shears greatly to spread large strain over the whole front hard phase, which deconcentrates stress at the crack tip.

We record the stress–strain curves of the three gels with and without precut crack (Fig. 3d). For both the soft gel and the hard gel, the precut cracks significantly reduce the critical strain for rupture. For the TPN gel, however, samples with and without precut crack have comparable critical strain. The toughness of the soft gel, the hard gel, and the TPN gel are 607 J/m², 1326 J/m², and 4202 J/m² (Fig. 3e). The TPN gel possesses the highest toughness, although the volume fraction of the tough hard phase significantly decreases in comparison with the hard gel. To reveal the source of this much enhanced toughness, we load and unload the uncut samples to the $\varepsilon_c$ for their corresponding precut gels (Fig. 3f). We separate the work of extension to dissipated work and elastic work, $W(\varepsilon_c) = W_D(\varepsilon_c) + W_E(\varepsilon_c)$. For the soft gel, the hysteresis is small, $W_D(\varepsilon_c)/W(\varepsilon_c) = 1\%$, and the toughness comes from the rupture of the long

polymer chains. For the hard gel, the hysteresis is pronounced, $W_D(\varepsilon_c)/W(\varepsilon_c) = 51\%$, and the toughness comes from both the dissipated work and elastic work. The former mainly comes from the rupture of the short-chain network, and the latter mainly comes from the rupture of the long-chain network. The synergy amplifies the toughness of the hard gel relative to the soft gel. For the TPN gel, the hysteresis is also pronounced, $W_D(\varepsilon_c)/W(\varepsilon_c) = 53\%$, and the toughness comes from both the dissipated work and elastic work. At the crack tip, the soft phase deconcentrates stress over the hard phase. This stress deconcentration further amplifies the toughness of the TPN gel relative to the hard gel.

**Mechanical analysis of TPNs.** The TPNs combine the merits of IPNs and composites, and break the longstanding elasticity-toughness conflict of soft materials through macroscopic structural design, which is markedly different from the newly developed molecular-scale crosslinking design[31–33]. The TPN gel demonstrates simultaneous improvement in elastic limit strain (seven times) and in toughness (three times) compared with the hard gel. The successful integration of high elasticity and high toughness needs to fulfill the following four requirements.

(i)   The soft phase needs to have high strength. The soft and hard segments in the stripe-patterned TPNs alternate in

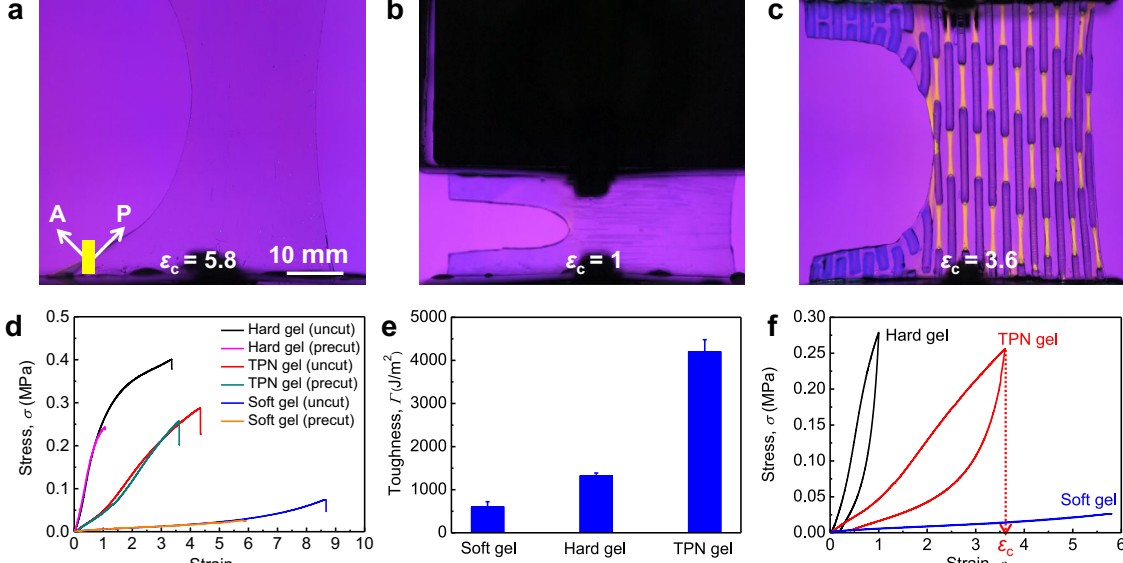

**Fig. 3 Fracture of TPNs. a–c** In situ polarizing optical observation of **a** soft gel, **b** hard gel, and **c** TPN gel. In each case, the image is taken when the gel with a precut crack is stretched to the critical strain $\varepsilon_c$ for rupture. **d** Stress–strain curves of gels with and without precut crack. **e** Toughness of the three gels. At least three samples were tested for the calculation of average value and standard deviation. **f** Load–unload stress–strain curves of the three uncut gels stretched to the critical strains for their corresponding precut gels.

series, and are subjected to the equal stress. The soft segment is strong enough to make the hard segment fracture preferentially (Supplementary Fig. 1c). The same is true in the staggered-patterned TPNs (Fig. 2a and Supplementary Movie 4). The strength of the fully swollen soft gel is far smaller than that of the hard gel (Fig. 3d). The restricted swelling, resulting from topological entanglement with the neighboring hard phase, imparts the soft phase with higher strength, which also can be confirmed by the remarkable birefringence phenomenon of the highly stretched soft phase, but not the soft gel of the same chemical composition even if stretched to the same extent (Fig. 3a, c). The restricted swelling of the soft phase depends on the interval between two neighboring hard phases, and disappears when the interval is too large (Supplementary Fig. 4).

(ii) The soft and hard phases must adhere strongly. Interfacial adhesion is the critical challenge for the design of composite materials. We attain strong interfacial adhesion by topological entanglement of polymer networks. The strong adhesion helps to smoothly transfer the stress between the phases. In all the mechanical measurements of TPNs, we have not observed any interfacial fracture.

(iii) The aspect ratio of the hard phase should be suitably chosen. We first consider two extreme cases about the aspect ratio $r$ of the hard phase. When $r = 0$, the TPN will regress to the soft gel, and lose high toughness. When $r = \infty$, the hard phase becomes continuous fibers, making the TPN lose high elastic limit strain. Therefore the TPNs must have an intermediate aspect ratio to reconcile the elastic limit strain and toughness. With the increase in $r$, the elastic limit strain decreases, whereas the toughness first increases to a peak value at $r = 4$, and then decreases slowly (Supplementary Fig. 5f). The case of $r = 4$ demonstrates superb combination of elastic limit strain ($\varepsilon_e = 1.5$) and toughness ($\Gamma = 4202$ J/m²). The decrease of $\varepsilon_e$ is attributed to the decreased volume fraction $\phi_s$ of the soft segment in the composite column when $r$ increases. The toughness $\Gamma$ is mainly determined by the energy dissipation in the process

zone, which is localized in the hard block at the crack front (Supplementary Fig. 5g). Write $\Gamma \sim W_h l_p$, where $W_h$ is the work of extension of the hard block and $l_p$ is the process zone size[12]. $l_p$ initially increases as the aspect ratio $r$ and then is saturated after exceeding a critical $r = 4$, which is responsible for the $r$-dependent change of the toughness.

(iv) The hard and soft phases should have large modulus contrast. The modulus of the hard phase has a great influence on the mechanical properties of TPNs, which can be easily tuned by changing the crosslinker concentration $C_{MBAA}$ for the short-chain network. We prepare a series of TPNs with the different modulus ratio $E_h/E_s$ of the hard phase to the soft phase, and find they have the similar critical strain $\varepsilon_c$ (Supplementary Fig. 6h–l). The elastic limit strain $\varepsilon_e$ and toughness $\Gamma$ deteriorate seriously when $E_h/E_s = 4.7$, and almost keep constant at a high level when $E_h/E_s \geq 9.4$ (Supplementary Fig. 6m). A large modulus ratio is indispensable to the high elasticity-toughness integration for the TPNs. At $E_h/E_s = 4.7$ (namely $C_{MBAA} = 2$ mol%), the low $\varepsilon_e$ is attributed to the large swelling of the hard segment and thus the decreased volume fraction $\phi_s$ of the soft segment (Supplementary Fig. 6a–e), while the low $\Gamma$ is attributed to the extremely small work of extension $W_h$ of the hard block at the crack front according to $\Gamma \sim W_h l_p$ (Supplementary Fig. 6n–p).

Except for elasticity and toughness, the stiffness and anti-fatigue of TPNs are also improved significantly. The modulus of the soft gel, the TPN gel, and the hard gel are 0.006 MPa, 0.04 MPa, and 0.2 MPa (Fig. 2d). The modulus of the TPN gel increases with the aspect ratio $r$ of the hard phase, in good agreement with the theoretical predictions from the tension-shear chain model[34] (Supplementary Fig. 7b). The analytic equation clearly indicates that the aspect ratio of the hard phase essentially compensates for the low stiffness of the soft phase through the product $r^2 E_s$ (Supplementary Fig. 7a). Under cyclic loads at the same maximum strain $\varepsilon_{max} = 0.5$, both the soft and hard gel are susceptible to fatigue fracture with a similar crack extension rate of 0.2 μm/cycle, but the TPN gel remains intact (Supplementary

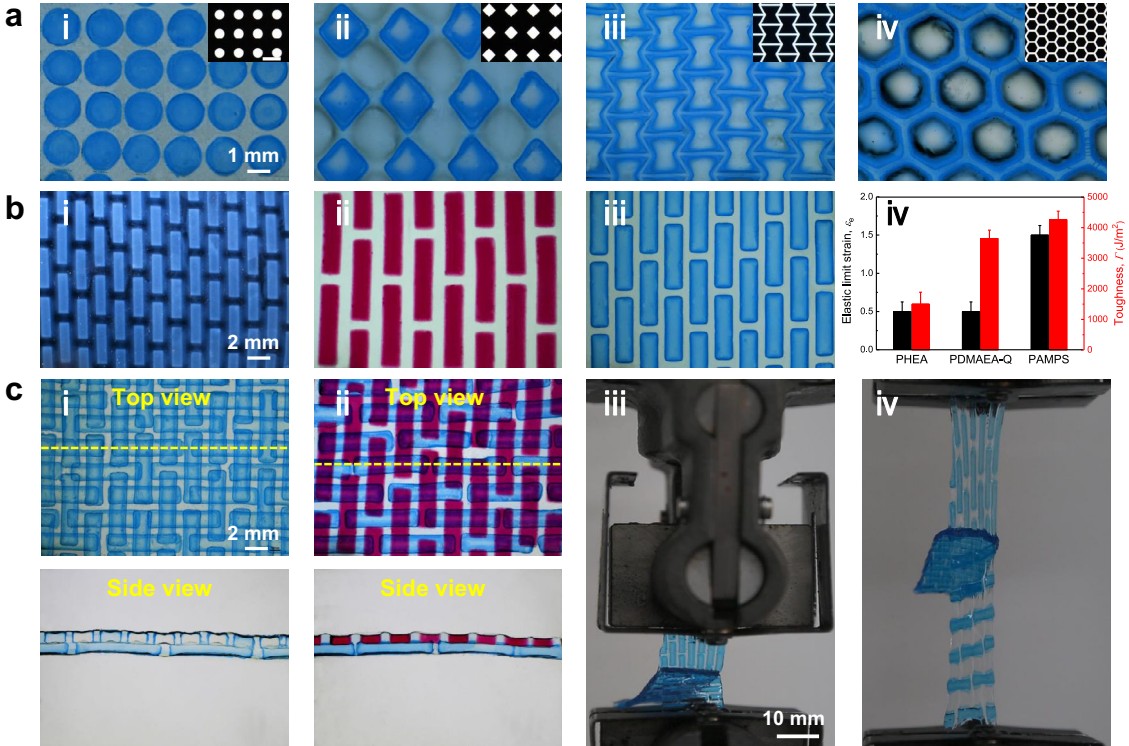

**Fig. 4 Extensibility of TPNs in geometric patterns, material combinations, and multilayers. a** Discrete (i, ii) and continuous (iii, iv) architectures in TPNs. The insets show the mask micrographs. **b** Micrographs of three representative TPNs in which the hard phases are made of neutral poly(hydroxyethyl acrylate) (PHEA) (i), cationic poly(*N,N*-dimethylamino ethylacrylate methyl chloride quarternary) (PDMAEA-Q) (ii), and anionic PAMPS (iii), along with their elastic limit strain and toughness (iv). At least three samples were tested for the calculation of average value and standard deviation. **c** Micrographs and peel test of bilayer TPNs with orthogonal staggered pattern. The bilayer is assembled by two identical PAMPS-based gels (i) or one PAMPS-based gel and the other PDMAEA-Q-based gel (ii). The yellow dashed lines represent the position of the cross-section for the side view. A crack is introduced at the interface of the bilayer (iii). When the bilayer is peeled, the two layers stretch and the crack does not advance (iv).

Fig. 8). These results strongly suggest the tremendous potential of TPN design principle for expanding the space of material properties.

**Extensibility of TPNs**. TPNs further expand the space of material properties by geometric patterns, material combinations, and multilayers (Fig. 4). TPNs of various patterns are readily fabricated by designing photolithographic masks (Fig. 4a). Polymer networks of various characteristics are available to construct TPNs. For example, we fabricate TPNs in which the hard phases are neutral, cationic, and anionic polymers, and find improvement in elastic limit strain and toughness compared with their corresponding hard gels (Fig. 4b). Furthermore, we fabricate bilayer TPNs by stacking two staggered-patterned scaffolds orthogonally and polymerizing the third network (Fig. 4c). In a peel test of the bilayer TPN gel with a crack at the interface between the layers, we observe cohesive fracture in the bulk gel, but not adhesive fracture at the interface (Supplementary Movie 5). This observation confirms that strong adhesion induced by topological entanglement also forms between the layers. The multilayer TPNs can potentially integrate diverse polymers to mimic the structures and functions of lamellar tissues, such as skins and blood vessels.

In summary, we have developed a class of polymer network morphology, the topoarchitected polymer networks (TPNs), which greatly expand the space of material properties. The TPNs integrate diverse polymers and patterns by sequential polymerization, photolithography, and stacking. The TPNs resolve a longstanding challenge in fabricating architected materials with strong adhesion through topological entanglement. As a

demonstration, we fabricate tendon-inspired TPNs that simultaneously achieve high elastic limit strain and high toughness. We further fabricate TPNs of various geometric patterns, material combinations, and multilayer stacks. We have fabricated TPNs using mask photolithography, but TPNs can also be fabricated using other methods, such as stereolithography[35]. In addition to hydrogels, the TPN principle also applies to other polymer materials, including plastics and elastomers. It is hoped that the TPN technology will be soon developed to enable breakthroughs in material properties.

## Methods

**Materials**. Unless otherwise mentioned, all chemicals were purchased from Aladdin and used without further purification. Monomers: Acrylamide (AAm, recrystallized from chloroform), 2-acrylamido-2-methylpropanesulfonic acid (AMPS), 2-hydroxyethyl acrylate (HEA), and *N,N*-dimethylamino ethylacrylate methyl chloride quarternary (DMAEA-Q, J&K Scientific). Crosslinker: *N,N'*-methylenebis(acrylamide) (MBAA). Initiators: 2-Oxoglutaric acid (OA), 2-hydroxy-2-methylpropiophenone (HMPP), and 2-hydroxy-4'-(2-hydroxyethoxy)-2-methylpropiophenone (Irgacure 2959). Dyes: Alcian blue and Amaranth. Patterned photomasks were designed by us and fabricated by GX Photomask (Shenzhen) Co. Ltd.

**Preparation of TPN hydrogels**. TPNs were prepared through three-step sequential polymerization and photolithography (Fig. 1e). The scaffold was made from an aqueous precursor solution, containing 2 M AAm, 0.01 mol% MBAA, and 0.01 mol% OA (mol% was relative to the AAm monomer concentration). The precursor was poured into a mold consisting of a 100 μm thick silicone spacer sandwiched by two parallel glass plates, and then polymerized in a nitrogen atmosphere with UV irradiation (365 nm, 2 mW/cm²) for 8 h. The as-prepared PAAm gels were leached in deionized water, and were used as scaffolds. We submerged the scaffolds in an aqueous precursor solution containing 1 M AMPS, 4 mol% MBAA, and 1 mol% HMPP to an equilibrium state. We sandwiched the

precursor-containing scaffolds between glass and polyethylene terephthalate (PET) release film. The PET film was covered by patterned masks and subjected to UV irradiation (365 nm, 120 mW/cm²) for several seconds in the air. The as-prepared PAAm/PAMPS gels were leached in deionized water to obtain patterned scaffolds. The patterned scaffolds were further immersed in the aqueous precursor solution containing 4 M AAm, 0.001 mol% MBAA, and 0.01 mol% OA to an equilibrium state, and then covered by two parallel glass plates. After the UV irradiation (365 nm, 2 mW/cm²) for 8 h in a nitrogen atmosphere, the TPN gels were leached in deionized water, and were ready for various characterization tests.

The patterned TPN gels consisted of soft phase (PAAm/PAAm) and hard phase (PAAm/PAMPS/PAAm). The modulus ratio of the hard phase to the soft phase was tuned by changing the MBAA concentration in the second precursor. As the control cases to the TPN gels, the soft gels were prepared without the second precursor, and the hard gels were prepared without photomasks. PHEA-based and PDMAEA-Q-based TPN gels were synthesized by replacing the PAMPS precursor. Note that the UV initiator Irgacure 2959 was used in the third precursor for PDMAEA-Q-based TPN gels. Bilayer TPN gels were obtained by stacking two layers of patterned scaffolds and then polymerizing the third PAAm network.

**Selective dyeing for TPN hydrogels**. To visualize the hard phase in TPN gels, the charged hard phase was selectively dyed by adsorbing dye molecules with the opposite charges (Supplementary Fig. 9). PAMPS-based TPN gels were immersed in 3 vol% acetic acid aqueous solution containing 1 wt% Alcian Blue for 15 min and then leached by deionized water. The tetravalent cationic Alcian Blue dyes the anionic PAMPS network selectively, but not the neutral PAAm matrix, which produces a sharp contrast between the hard phase and the soft phase. Similarly, PDMAEA-Q-based TPN gels were immersed in 0.1 wt% Amaranth aqueous solution for 15 min and then leached by deionized water.

**Modulus ratio characterization**. A tensile test was performed with a commercial testing machine (Shimadzu AGS-X). Fully swollen stripe-patterned TPN gels were cut into a dumbbell shape by a standardized cutter with width of 2 mm and height of 12 mm. Both ends of the dumbbell-shaped samples were clamped and stretched at a constant velocity of 100 mm/min, by which the stress–strain curves were recorded. The tensile process of the stripe-patterned gels obeys the series model (isostress model), so the modulus ratio of the hard phase to the soft phase equals to the inverse ratio of their strains ($E_h/E_s = \varepsilon_s/\varepsilon_h$) in the elastic tensile region. The strain ratio $\varepsilon_s/\varepsilon_h$ was estimated through the in situ observation of deformation (Supplementary Fig. 6f).

**Elasticity characterization**. To include enough hard phase, the TPNs were cut into a dumbbell shape by a nonstandardized cutter with width 10 mm and height 20 mm. The dumbbell-shaped samples were gripped and stretched at a constant velocity of 100 mm/min. The whole tensile process was recorded by a homemade polarizing system including all the constituents as shown in Supplementary Fig. 2a. The strain $\varepsilon_s$ and $\varepsilon_h$ of the soft and hard segments locating in the composite column were estimated through quantitative image analysis. The elasticity of gel samples was measured by load–unload cycle test with the increasing maximum strain $\varepsilon_{max}$. For each cycle, the area under the load curve defines the work of extension $W$, the area between the load and unload curves defines the dissipated work $W_D$, and the area under the unload curve defines the elastic work $W_E$ (Fig. 2e). The hysteresis was characterized by the ratio $W_D/W$ as a function of $\varepsilon_{max}$. The onset strain where hysteresis started to increase abruptly was defined as elastic limit strain $\varepsilon_e$.

**Toughness characterization**. The toughness of gels was defined as the energy consumed per unit area growth of cracks and was characterized by a pure shear test. A precut sample with a rectangular geometry (width 50 mm and height $H = 10$ mm) and a 20 mm one-edge crack were used. The sample was clamped and stretched along the $H$ direction at a constant velocity of 30 mm/min, by which the stress–strain curve was recorded. The strain at which the precut crack started to propagate was defined as the critical strain $\varepsilon_c$, and the integrated area under the stress–strain curve from 0 to $\varepsilon_c$ was defined as the critical work of extension $W(\varepsilon_c)$. The toughness was given by $\Gamma = W(\varepsilon_c)H$. Note that the stress was corrected by the initial effective sample width of 30 mm. With this correction, the stress–strain curve of the precut sample coincided with that of the uncut sample with the same geometry in the range of $0 \leq \varepsilon \leq \varepsilon_c$ (Fig. 3d), so the uncut sample used in the traditional pure shear test[14] was no longer needed in our modified method.

**Finite element simulation**. We used the commercial finite element software, ABAQUS, to calculate the strain distribution of TPNs under uniaxial tension. The simulated geometry was set as a thin sheet (width 15 mm and height 30 mm) including seven columns of hard phase (width 1.5 mm and height 6 mm) (Supplementary Fig. 3). We used the Neo-Hookean model to describe the stress–strain behavior of TPNs. The hydrogels were considered as incompressible, so only two parameters, shear modulus of the soft and hard phases, needed as inputs, which were identified from experimental measurements. We chose the CPS4R element in ABAQUS. The sample was stretched to $\varepsilon = 1$, and the strain distribution was calculated.

## Data availability

The authors declare that the data supporting the findings of this study are available within the paper and its Supplementary Information files or from the corresponding authors on reasonable request.

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

## Acknowledgements
J.H. acknowledges the support from the National Natural Science Foundation of China (11702207). Z.S. acknowledges the support from the Harvard University Materials Research Science and Engineering Center (DMR-2011754). T.L. acknowledges the support from the National Natural Science Foundation of China (11922210). Z.W. acknowledges the support from the National Natural Science Foundation of China (12002255).

## Author contributions
Z.S. and J.H. conceived the idea. X.L., Z.S., and J.H. designed the research. X.L. performed most of the experiments. J.W. performed the polarizing device. K.Q. performed the restricted swelling measurement. G.L. and Z.W. performed the finite element simulation. T.L. participated in the mechanical analysis. X.L., Z.S., and J.H. analyzed the data and wrote the manuscript with the input from all authors.

## Competing interests
The authors declare no competing interests.
