## [Peer Review File · Nature Communications]

Topoarchitected polymer networks expand the space of material propertiesReviewers' comments:

Reviewer #1 (Remarks to the Author):

The manuscript titled "Topoarchitected polymer networks expand the space of material properties" by Liu et al explores an innovative fabrication process to replicate the molecular toughening mechanism of double networks at macroscales to enhance toughness and modulus at the same time. The authors utilised a three-step polymerisation process to create orderly heterogeneous hydrogel constructs made of lightly crosslinked matrices and highly crosslinked hard blocks. The staggered architecture was obtained by in-situ polymerisation of the hard segments within the first loosely crosslinked matrix using patterned masks followed by further polymerisation of a third network. The approach taken in this work is very interesting and can be easily adapted for other sets of materials (e.g. organogels, elastomers), and other fabrication techniques. In general, the marriage between toughness and elasticity – as pointed by the authors – is hard to achieve in swollen networks because of competing factors that act against each other. The authors have been able to achieve this by practically implementing the toughening mechanism of double networks at a larger scale. Given the ease of implementation and higher degrees of freedom, one can see that this method could be also used to create materials with gradient mechanical properties. Other functional materials (e.g. stimuli-responsive hydrogels) can be incorporated into the system to make soft actuators etc. The results presented in this work are noteworthy, original, and significant for a wide audience. The experimental data is described well, and methods are clearly outlined. The conclusion is supported by experimental data and discussion is sound and complete.

Here are some suggestions for the authors to consider:

- 1) The interaction between hard segments and the matrix is suggested to have a physical nature (entanglement). It is however possible to have chemical bonding between all the three elements given the nature of radical polymerisation. Can you please comment on that?
- 2) Do you expect to see the same outcome if you create the staggered pattern first by polymerisation the hard segments (patterned) and then polymerising the other networks threading the hard segments? If topology the way described in this work is in play one should expect to see similar outcomes regardless of the order of fabrication (?)
- 3) You have individually presented some relations between aspect ratio (r) of hard segments and moduli ratios. Is it possible to use mechanics of composites to theoretically model the stiffness of these gels (e.g. fibre reinforced composites)? (Or perhaps the interlocking between the networks will deviate from the theory.)

Reviewer #2 (Remarks to the Author):

This review is in regard to the article "Topoarchitected polymer networks expand the space of material properties" as submitted to Nature Communications. I read this article in full, carefully evaluated all information provided, and have no conflict of interest. As to my evaluation: This is a very well-done study that explores the effect of designing heterogeneity of a double network with the goal of maximizing "elasticity" or critical strain and toughness. Its main motivation is to expand our material horizon to materials that can do both: be highly stretchable while also resisting fracture. The article is well written and uses illustrations to communicate methods and results. It is clear to the reviewer that the authors have done a large amount of work and have carefully designed this study. Despite the intense efforts, the reviewer has a number of questions and concerns, some minor, some major:

Major:

[1] While the reviewer appreciates that the article is motivated by a naturally occurring material, in this case tendon, it seems somewhat forced. Specifically, tendon, in fact, does NOT provide high elasticity with a failure strain of ~10-20%. Additionally, the scale of heterogeneities varies drastically between tendon and collagen. I think the article may be better served by choosing another motivating material or referring to a broader class of materials rather than choosing one, imperfect example.

[2] The introduction discusses tough and soft materials as if there wasn't a very large body of literature on "tough hydrogels". This came as surprise to the reviewer given that the authors have themselves significantly contributed to this body of work. In fact, the word "tough hydrogel" does not appear a single time in the manuscript. Moreover, the existence of a large body of other materials that already do what the authors try to achieve somewhat dampens the enthusiasm about a material that may potential not be superior to those materials while being more complex to manufacture. Similarly, the word "elastic dissipator" is nowhere to be mentioned while their material shares many of the features and is essentially motivated by the same goals and mechanisms.

[3] One other potential oversight is the authors not demonstrating that it is, indeed, the architecture of the material that lends the material its toughness? In the end, the material is a double network hydrogel. While of course, with a well-designed structure, the review can't help but wonder how this material would compare to a disorganized hydrogel of the same networks without the architected elements? That is, the authors compare their optimal material to the single network materials as well as the their architected material with different stiffness and geometric parameters. However, they don't compare their material to a disorganized double network of the same constituents, mass fractions etc? Is this not practically possible?

[4] Figure S3 is unclear. It'd be helpful to also show the undeformed mesh where the different regions are highlighted in color. Also, it is not clear what is shown: strain? Stress? If so, what strain? What stress? And it is also not clear why the principal values are shown rather than, say, the shear strains/stresses given that it is the authors intention to demonstrate their existence?

[5] The authors illustrate in Figure S4 prestrain or internal strain as a result of different swelling between networks. What is the effect of this prestrain on the internal mechanics? Are those effects included in the finite element simulation? What are the authors anticipated effect of this swelling-induced residual stress?

[6] In the same vein as [5], overall, the investigation and discussion are rather phenomenological. That is, there are no mechanistic explanations as to why certain ratios in stiffness or certain geometric parameters are superior to others. Rather, a large parameter space is experimental swept and conclusions are drawn empirically. After reading the article, I am left with many questions as to why certain parameters matter and others don't.

[7] While the authors argue that the stiffness ratio between the matrix and the hard elements matters, Figure S6(m) seems to indicate that, beyond a value of 5, both the elastic limit and the toughness are pretty insensitive to the stiffness ratio. As highlighted in [6], I wished there was more of a mechanistic analysis conducted that would explain why?

[8] Figure 4 was insufficiently labelled for the review. Additionally, the caption was insufficient to fully understand this very busy figure.

[9] The authors calculate toughness based on a very simple formula first explored by Rivlin and Thomas in the 1950s. They originally proposed this formula for a material with negligible hysteresis. By ignoring other dissipative phenomena, the authors inadvertently couple changes in the hysteresis behavior of their various material iterations with its "intrinsic fracture toughness". Is this the best way to go about this analysis? Are there alternatives?

[10] In the discussion the author postulate the requirements for a material with high elasticity and high toughness. However, some of these requirements remain untested. For example, nowhere do the authors specifically test whether it is, indeed, the adhesion between the networks that drives their material's superior properties. Is there a control experiment in which the architecture could be maintained but the adhesion between the networks reduced or removed?

[11] Finally, and most importantly. As highlighted in [2], there have been many materials that are very

extensible, while being tough. However, as the authors know, many of them are highly susceptible to fatigue. The “new” challenge in this material community has been to design materials that are extensible, tough, and fatigue-resistant. It would be exciting to expand the testing of their materials to fatigue.

Reviewer #1 (Remarks to the Author):

The manuscript titled “Topoarchitected polymer networks expand the space of material properties” by Liu et al explores an innovative fabrication process to replicate the molecular toughening mechanism of double networks at macroscales to enhance toughness and modulus at the same time. The authors utilised a three-step polymerisation process to create orderly heterogeneous hydrogel constructs made of lightly crosslinked matrices and highly crosslinked hard blocks. The staggered architecture was obtained by in-situ polymerisation of the hard segments within the first loosely crosslinked matrix using patterned masks followed by further polymerisation of a third network. The approach taken in this work is very interesting and can be easily adapted for other sets of materials (e.g. organogels, elastomers), and other fabrication techniques. In general, the marriage between toughness and elasticity – as pointed by the authors – is hard to achieve in swollen networks because of competing factors that act against each other. The authors have been able to achieve this by practically implementing the toughening mechanism of double networks at a larger scale. Given the ease of implementation and higher degrees of freedom, one can see that this method could be also used to create materials with gradient mechanical properties. Other functional materials (e.g. stimuli-responsive hydrogels) can be incorporated into the system to make soft actuators etc.

The results presented in this work are noteworthy, original, and significant for a wide audience. The experimental data is described well, and methods are clearly outlined. The conclusion is supported by experimental data and discussion is sound and complete.

Response: We thank the reviewer for the positive evaluation and the constructive comments on the manuscript. We address the comments below.

Here are some suggestions for the authors to consider:

1) The interaction between hard segments and the matrix is suggested to have a physical nature (entanglement). It is however possible to have chemical bonding between all the three elements given the nature of radical polymerisation. Can you please comment on that?

Response: The reviewer is right. In the classic poly(2-acrylamido-2-methylpropanesulfonic acid) (PAMPS)/polyacrylamide (PAAm) double network (DN) hydrogels, J. P. Gong et al. confirmed the existence of the internetwork chemical bonding due to the residual double bonds from the crosslinker in the first PAMPS network (*J. P. Gong et al., Macromolecules 2009, 42, 2184*). They also concluded the internetwork chemical bonding is equivalent to extra crosslinkers for the second PAAm network. Even if in an initiator-treated DN gel without internetwork chemical bonding, high strength and toughness are still well maintained. These previous investigations strongly suggest physical entanglement between the two networks is dominant for force transfer.

In our topoarchitected polymer network (TPN) gels containing PAAm/PAMPS/PAAm triple network, the case is similar to DN gels. There should be internetwork chemical bonding especially between the second PAMPS network and the third PAAm network, given that the irradiation time for the PAMPS polymerization is only 8 s. However, the physical entanglement between the three networks is the main factor responsible for force transfer and interfacial adhesion of TPN gels. The entanglement effect has been widely adopted to improve hydrogels' mechanical properties, such as our previous microgel-reinforced hydrogels (*J. Hu et al.,*

Macromolecules 2011, 44, 7775) and the newest highly entangled hydrogels (Z. G. Suo *et al.*, *Science* 2021, 374, 212).

2) Do you expect to see the same outcome if you create the staggered pattern first by polymerisation the hard segments (patterned) and then polymerising the other networks threading the hard segments? If topology the way described in this work is in play one should expect to see similar outcomes regardless of the order of fabrication?

Response: It is indeed feasible in principle as the reviewer considers, but not in the specific operation. When the staggered pattern is formed first, it will be immersed into the acrylamide precursor solution to polymerize the matrix network. The discrete pattern cannot be maintained during this swelling process. In our three-step sequential polymerization, the first PAAm network just serves as the scaffold to host the staggered pattern, but not contributes much to the final mechanical performances of the TPN hydrogels.

3) You have individually presented some relations between aspect ratio (r) of hard segments and moduli ratios. Is it possible to use mechanics of composites to theoretically model the stiffness of these gels (e.g. fibre reinforced composites)? (Or perhaps the interlocking between the networks will deviate from the theory.)

Response: Based on a tension-shear chain model, H. J. Gao *et al.* gave an analytic equation to describe the effective modulus of biocomposites with staggered mineral inclusions embedded in protein matrix (H. J. Gao *et al.*, *J. Mech. Phys. Solids* 2004, 52, 1963). Considering the similar structure between TPN gels and biocomposites, we attempt to use the tension-shear chain model to theoretically predict the stiffness of TPN gels. When changing the aspect ratio of the hard phase r and the modulus ratio E_h/E_s , the experimental values are in good agreement with the theoretical predictions.

We have added the discussion on the stiffness of TPN gels in the main text on **Page 12** and a figure in the supplementary information as Fig. S7.

“Except for elasticity and toughness, the stiffness and anti-fatigue of TPNs are also improved significantly. The modulus of the soft gel, the TPN gel, and the hard gel are 0.006 MPa, 0.04 MPa, and 0.2 MPa (Fig. 2d). The modulus of the TPN gel increases with the aspect ratio r of the hard phase, in good agreement with the theoretical predictions from the tension-shear chain model³¹ (Supplementary Fig. S7b). The analytic equation clearly indicates that the aspect ratio of the hard phase essentially compensates for the low stiffness of the soft phase through the product r^2E_s (Supplementary Fig. S7a).”

Figure S7 | Comparison of the modulus of TPNs predicted by the tension-shear chain model and by experimental measurement. a, A tension-shear chain model of TPNs in which the tensile zones of the soft phase are eliminated to emphasize the load transfer within the composite structure. The analytic equation for the modulus of TPNs refers to the previous report², including four parameters as E_h , E_s , ϕ , and r . **b**, E/E_h as a function of the aspect ratio r of the hard phase. The experimental values are in good agreement with the theoretical predictions from the tension-shear chain model. Note that the experimental E/E_h at $r = \infty$ is approximately plotted at $r = 30$, because the tension-shear chain model predicts the E/E_h nearly keeps constant when $r \geq 30$. **c**, E/E_h as a function of the modulus ratio E_h/E_s of the hard phase to the soft phase. When changing the crosslinker concentration for the PAMPS network, ϕ and E_h/E_s vary simultaneously, but r keeps constant. The experimental values are close to the theoretical predictions.

Reviewer #2 (Remarks to the Author):

This review is in regard to the article “Topoarchitected polymer networks expand the space of material properties” as submitted to Nature Communications. I read this article in full, carefully evaluated all information provided, and have no conflict of interest. As to my evaluation: This is a very well-done study that explores the effect of designing heterogeneity of a double network with the goal of maximizing “elasticity” or critical strain and toughness. Its main motivation is to expand our material horizon to materials that can do both: be highly stretchable while also resisting fracture. The article is well written and uses illustrations to communicate methods and results. It is clear to the reviewer that the authors have done a large amount of work and have carefully designed this study. Despite the intense efforts, the reviewer has a number of questions and concerns, some minor, some major:

Response: We are thankful for the reviewer’s time and efforts as well as the constructive comments on the manuscript. We address the comments below.

Major:

[1] While the reviewer appreciates that the article is motivated by a naturally occurring material, in this case tendon, it seems somewhat forced. Specifically, tendon, in fact, does NOT provide high elasticity with a failure strain of ~10-20%. Additionally, the scale of heterogeneities varies drastically between tendon and collagen. I think the article may be better served by choosing another motivating material or referring to a broader class of materials rather than choosing one, imperfect example.

Response: Tendons are typically elastic and tough soft tissues with complex multiscale hierarchical structure. People have revealed the excellent mechanical properties of tendons dominantly dependent on the staggered alignment of collagen fibrils in a proteoglycan matrix at the scale of fascicles (*H. J. Gao et al., J. Mech. Phys. Solids 2004, 52, 1963*). The previous experimental observations have also indicated the proteoglycan matrix is subjected to considerable deformation during the extension process of tendons (*J. E. Scott et al., J. Anat. 1995, 187, 423*). Inspired from the structure-property relationship of tendons, we therefore design highly elastic and tough TPN hydrogels by embedding staggered architecture of hard blocks in a soft matrix. Tendons truly provide a small failure strain of ~15%, but we would like to learn the deformation and toughening mechanism from tendons, rather than fabricate materials to replace them.

We have made a revision in the main text on **Page 5**.

“Although the Achilles tendon has a modest rupture strain of ~15%, the proteoglycan matrix is subjected to considerable deformation during the extension process²⁸.

Learning from the structure-property relationship of the Achilles tendon, here we demonstrate that a similarly architected material can achieve both high elastic limit strain and high toughness.”

[2] The introduction discusses tough and soft materials as if there wasn’t a very large body of literature on “tough hydrogels”. This came as surprise to the reviewer given that the authors

have themselves significantly contributed to this body of work. In fact, the word “tough hydrogel” does not appear a single time in the manuscript. Moreover, the existence of a large body of other materials that already do what the authors try to achieve somewhat dampens the enthusiasm about a material that may potential not be superior to those materials while being more complex to manufacture. Similarly, the word “elastic dissipator” is nowhere to be mentioned while their material shares many of the features and is essentially motivated by the same goals and mechanisms.

Response: We define the elastic limit strain by the maximum strain at which the load and unload curves coincide (Fig. 2e). The elastic limit strain is used to characterize the elasticity of materials, but not the common fracture strain. The studies on tough and stretchable hydrogels indeed have lasted for two decades, but there are few reports solving the elasticity-toughness conflict (Fig. 1a). In fact, we first proposed the significance for the elasticity-toughness conflict just in 2019 on our PNAS paper (Z. G. Suo *et al.*, *PNAS* 2019, 116, 5967). It is a new research topic.

We have revealed that the toughness of TPN hydrogels comes from the comparable contribution of the dissipated work and elastic work. The former is attributed to the bond rupture of the PAMPS network, and the latter to the extension of the PAAm networks, namely “*elastic dissipator*” as the reviewer mentioned.

We have added the definition of elastic limit strain and toughness on **Page 3**, and highlighted the elastic limit strain in Fig. 2e by a red dotted arrow.

“The difficulty in simultaneously achieving elasticity and toughness is evident on the plane of elastic limit strain (ϵ_e , the maximum strain at which the load and unload curves coincide) and toughness (T , the energy consumed per unit area during crack propagation) (Fig. 1a).”

[3] *One other potential oversight is the authors not demonstrating that it is, indeed, the architecture of the material that lends the material its toughness? In the end, the material is a double network hydrogel. While of course, with a well-designed structure, the review can't help but wonder how this material would compare to a disorganized hydrogel of the same networks without the architected elements? That is, the authors compare their optimal material to the single network materials as well as the their architected material with different stiffness and geometric parameters. However, they don't compare their material to a disorganized double network of the same constituents, mass fractions etc? Is this not practically possible?*

Response: We conducted the comparison as the reviewer mentioned. The hard gel in Fig. 3 has the triple network as the hard phase in TPN gel, but not the single network. It is the disorganized hydrogel of the same networks without the architected elements. Fig. 3e clearly indicates the toughness increases from 1326 J/m² to 4202 J/m² after the introduction of the staggered architecture, which strongly demonstrates the effect of the TPN architecture on the mechanical properties. The description about the soft gel and hard gel was written as “*Also plotted are the stress-strain curves of the soft gel and hard gel, which have the same chemical composition as the soft and hard phases in the TPN.*” on **Page 8**.

[4] *Figure S3 is unclear. It'd be helpful to also show the undeformed mesh where the different regions are highlighted in color. Also, it is not clear what is shown: strain? Stress? If so, what*

strain? What stress? And it is also not clear why the principal values are shown rather than, say, the shear strains/stresses given that it is the authors intention to demonstrate their existence?

Response: Thank you for your good suggestions. We have added the undeformed mesh and highlighted the hard phase by yellow dotted outlines in Fig. S3. In Fig. S3, we intend to use finite element simulation to show the shear strain distribution during the tension of TPNs. Indeed, it is clearer to show the shear strain than the principal strain.

We have revised Fig. S3 and its caption as the reviewer suggested.

Figure S3 | Finite element simulation for the shear strain distribution during the tension of TPNs. When the sample is stretched to $\epsilon = 1$, the strain distribution shows complex tension-shear coupling strain in the soft column and pure tensile strain in the composite column.

[5] The authors illustrate in Figure S4 prestrain or internal strain as a result of different swelling between networks. What is the effect of this prestrain on the internal mechanics? Are those effects included in the finite element simulation? What are the authors anticipated effect of this swelling-induced residual stress?

Response: Prior to mechanical tests, all the gel materials are fully swollen in water until equilibrium. If without restriction, the lowly crosslinked PAAm network will swell to a large extent, resulting in the decreased chain density and mechanical properties. Fig. S4 is to explain why the strength of the soft phase in the TPN gel is higher than that of the soft gel. The highly

crosslinked PAMPS network (hard blocks) restricts the swelling of the PAAm network through topological entanglement. When the interval between two neighboring hard blocks increases, the restriction ability decreases greatly according to the volume expansion of the soft phase.

Therefore, the prestrain, corresponding to a larger swelling degree of the soft phase, should lead to poor mechanical properties of the TPN gel, including stiffness, strength, and toughness. In the TPN gel, the hard phase is the same as the hard gel, but the soft phase is different from the soft gel due to the restricted swelling. Considering this effect, the shear modulus of the soft and hard phases was input in the finite element simulation. When the interval between two neighboring hard blocks is large enough, the soft phase should be equivalent to the soft gel, and occur preferential fracture in comparison with the hard phase. The resultant TPN gel will deteriorate in the stiffness, strength, and toughness.

[6] In the same vein as [5], overall, the investigation and discussion are rather phenomenological. That is, there are no mechanistic explanations as to why certain ratios in stiffness or certain geometric parameters are superior to others. Rather, a large parameter space is experimental swept and conclusions are drawn empirically. After reading the article, I am left with many questions as to why certain parameters matter and others don't.

Response: With the increase in the aspect ratio r , the elastic limit strain ε_e decreases, whereas the toughness Γ first increases to a peak value at $r = 4$, and then decreases slowly (Supplementary Fig. S5f). We have concluded the soft segment in the composite column provides a dominant contribution to the deformation of TPNs (Fig. 2), so the decrease of the volume fraction of the soft segment upon r leads to the decrease of the elastic limit strain. The toughness Γ is mainly determined by the energy dissipation in the process zone, which is localized in the hard block at the crack front (Supplementary Fig. S5g). Write $\Gamma \sim W_h l_p$, where W_h is the work of extension of the hard phase and l_p is the process zone size (Z. G. Suo et al., PNAS 2019, 116, 5967). l_p initially increases as the aspect ratio r and then is saturated after exceeding a critical $r = 4$, which is responsible for the r -dependent change of the toughness.

We have added a new Fig. S5g and the mechanistic explanations on the effect of the aspect ratio on **Page 11**.

“The decrease of ε_e is attributed to the decreased volume fraction ϕ_s of the soft segment in the composite column when r increases. The toughness Γ is mainly determined by the energy dissipation in the process zone, which is localized in the hard block at the crack front (Supplementary Fig. S5g). Write $\Gamma \sim W_h l_p$, where W_h is the work of extension of the hard block and l_p is the process zone size¹². l_p initially increases as the aspect ratio r and then is saturated after exceeding a critical $r = 4$, which is responsible for the r -dependent change of the toughness.”

Figure S5 | Effect of the aspect ratio of the hard phase on the mechanical properties of TPNS. **a-e**, Optical micrographs of TPNS with varied aspect ratio r of the hard phase. For simplicity, we only change the length of the hard phase and keep the other geometric parameters constant. **f**, Effect of the aspect ratio on the elastic limit strain and toughness of TPNS. **g**, Process zone localized in the hard block at the crack front. The process zone size l_p initially increases as the aspect ratio r and then is saturated after exceeding a certain critical r . The toughness Γ of TPNS is mainly determined by the energy dissipation in the process zone. Write $\Gamma \sim W_h l_p$, where W_h is the work of extension of the hard block and can be measured from the tensile stress-strain curve of the hard gel with the same chemical composition.

[7] While the authors argue that the stiffness ratio between the matrix and the hard elements matters, Figure S6(m) seems to indicate that, beyond a value of 5, both the elastic limit and the toughness are pretty insensitive to the stiffness ratio. As highlighted in [6], I wished there was more of a mechanistic analysis conducted that would explain why?

Response: The modulus of the hard phase is controlled by changing the crosslinker concentration C_{MBAA} for the PAMPS network. At $E_h/E_s = 4.7$ (namely $C_{MBAA} = 2$ mol%), the low elastic limit strain ϵ_e is attributed to the large swelling of the hard segment and thus the decreased volume fraction ϕ_s of the soft segment (Supplementary Fig. S6a-e), while the low toughness Γ is attributed to the extremely small work of extension W_h of the hard block at the crack front according to $\Gamma \sim W_h l_p$ (Supplementary Fig. S6n-p). Combining Comments [6] and [7], we have conducted a qualitative mechanistic analysis that the process zone size l_p plays a dominant role on Γ when the aspect ratio r changes, while the work of extension W_h of the hard block plays a dominant role on Γ when the modulus ratio E_h/E_s changes.

We have added a new Fig. S6n-p and the mechanistic explanations on the effect of the modulus ratio on **Page 12**.

“At $E_h/E_s = 4.7$ (namely $C_{MBAA} = 2$ mol%), the low ε_e is attributed to the large swelling of the hard segment and thus the decreased volume fraction ϕ_s of the soft segment (Supplementary Fig. S6a-e), while the low Γ is attributed to the extremely small work of extension W_h of the hard block at the crack front according to $\Gamma \sim W_h l_p$ (Supplementary Fig. S6n-p).”

Figure S6 | Effect of the modulus ratio on the mechanical properties of TPNs. a-e, Optical micrographs of TPNs with varied crosslinker concentration C_{MBAA} for the second network. f, Strain ε_s and ε_h of the soft and hard phases in a stripe-patterned TPN as a function of the applied strain ε in the linear elastic region. The stripe-patterned TPN is obtained by cutting the overlap

region of the staggered-patterned TPN, marked by the red box in (b). The tensile process of the stripe-patterned TPN obeys series model (isostress model), so the modulus ratio of the hard phase to the soft phase $E_h/E_s = \varepsilon_s/\varepsilon_h = k_s/k_h$, here k means the slope of the fitting line. **g**, Dependence of E_h/E_s on C_{MBAA} . **h-i**, In-situ polarizing optical observation of various precut TPNs at the critical strain ε_c . **m**, Effect of E_h/E_s on the elastic limit strain and toughness of TPNs. **n**, Stress-strain curves of the hard gels with varied C_{MBAA} to approximately reflect the mechanical behaviors of the hard phase in TPNs. As the increase of C_{MBAA} , the hard gels show two types of stress-strain response, with yielding (2 mol%, 4 mol%, and 6 mol%) and without yielding (8 mol% and 10 mol%). **o**, Work of extension W_h of the hard block at the crack front of TPNs estimated by integrating the stress-strain curves of the hard gels. In the case with yielding, W_h is estimated by integrating to the yielding point, because the hard block at the crack front once yields, will break immediately without a whole yielding process present in the tension of the hard gel. In the case without yielding, W_h is estimated by integrating to the breaking point. **p**, Dependence of W_h on C_{MBAA} .

[8] Figure 4 was insufficiently labelled for the review. Additionally, the caption was insufficient to fully understand this very busy figure.

Response: Thank you for your careful review. We have labelled each panel in Fig. 4 and revised the caption to make it more readable.

We have revised Fig. 4 and its caption as follows.

Figure 4 | Extensibility of TPNs in geometric patterns, material combinations, and multilayers. **a**, Discrete (i, ii) and continuous (iii, iv) architectures in TPNs. The insets show the mask micrographs. **b**, Micrographs of three representative TPNs in which the hard phases are made of neutral poly(hydroxyethyl acrylate) (PHEA) (i), cationic poly(*N,N*-dimethylamino

ethylacrylate methyl chloride quarternary) (PDMAEA-Q) (ii), and anionic PAMPS (iii), along with their elastic limit strain and toughness (iv). **c**, Micrographs and peel test of bilayer TPNs with orthogonal staggered pattern. The bilayer is assembled by two identical PAMPS-based gels (i) or one PAMPS-based gel and the other PDMAEA-Q-based gel (ii). The yellow dashed lines represent the position of the cross section for side view. A crack is introduced at the interface of the bilayer (iii). When the bilayer is peeled, the two layers stretch and the crack does not advance (iv).

[9] The authors calculate toughness based on a very simple formula first explored by Rivlin and Thomas in the 1950s. They originally proposed this formula for a material with negligible hysteresis. By ignoring other dissipative phenomena, the authors inadvertently couple changes in the hysteresis behavior of their various material iterations with its “intrinsic fracture toughness”. Is this the best way to go about this analysis? Are there alternatives?

Response: In our previous work, we used pure shear test proposed by Rivlin and Thomas (*R. S. Rivlin & A. G. Thomas, J. Polym. Sci. 1953, 10, 291*) to measure the toughness of tough alginate-Ca/PAAm hydrogels, and verified the measured values are in good agreement with that obtained from two other methods as tensile test and double peel test (*Z. G. Suo et al., Nature 2012, 489, 133*). The pure shear test has been a widely adopted method for the studies on tough gels with large hysteresis. In this case, the measured toughness includes two contributions of dissipated energy and elastic energy.

[10] In the discussion the author postulate the requirements for a material with high elasticity and high toughness. However, some of these requirements remain untested. For example, nowhere do the authors specifically test whether it is, indeed, the adhesion between the networks that drives their material’s superior properties. Is there a control experiment in which the architecture could be maintained but the adhesion between the networks reduced or removed?

Response: In TPN hydrogels, the interfacial adhesion is formed on the basis of internetwork topological entanglement, which is achieved through monomer diffusion and in-situ polymerization. Therefore, in experiments, it is difficult to fabricate TPN hydrogels without adhesion or with weak adhesion. In theory, strong interfacial adhesion helps to transfer the load from the soft phase to the hard phase, and provides TPN composites better load-bearing capacity and higher energy dissipation; in contrary, weak adhesion will lead to preferential fracture failure at interface. The lower bound of the strength of the soft phase, hard phase, and interface determines the fracture mode of TPNs. We observed the preferential fracture always occurs in the hard phase of TPNs, indicating the interfacial adhesion is strong enough. Though weak adhesion is difficult to complete in TPN hydrogels as control, it was successfully designed in spandex fiber/PDMS composites in our previous work, clearly suggesting the interfacial strength governs the mechanical properties of fiber-reinforced composites (*Z. G. Suo et al., PNAS 2019, 116, 5967*).

[11] Finally, and most importantly. As highlighted in [2], there have been many materials that are very extensible, while being tough. However, as the authors know, many of them are highly susceptible to fatigue. The “new” challenge in this material community has been to design materials that are extensible, tough, and fatigue-resistant. It would be exciting to expand the testing of their materials to fatigue.

Response: Thank you for your good suggestions. Indeed, the studies on the fatigue of soft materials have developed rapidly in recent years. We have added the fatigue test for the soft, hard, and TPN gel with precut crack, and found the TPN gel shows the significantly improved fatigue-resistant property in comparison with the soft and hard gel.

We have added a new Fig. S8 and the related discussion about fatigue behaviors of the precut gel samples on *Page 12*.

“Under cyclic loads at the same maximum strain $\epsilon_{\max} = 0.5$, both the soft and hard gel are susceptible to fatigue fracture with a similar crack extension rate of $0.2 \mu\text{m}/\text{cycle}$, but the TPN gel remains intact (Supplementary Fig. S8). These results strongly suggest the tremendous potential of TPN design principle for expanding the space of material properties.”

Figure S8 | Fatigue test for precut samples. The soft, hard, and TPN gel with a rectangular geometry (width 50 mm and height 10 mm) and a 20 mm one-edge crack, were subjected to cyclic loads over 5000 cycles at the maximum strain of 0.5. The loading frequency was fixed at 1 Hz. The crack extension was monitored by a video camera (Nikon 5200).

REVIEWERS' COMMENTS

Reviewer #1 (Remarks to the Author):

Thank you for considering all the comments and suggestions provided. The manuscript has been revised accordingly and new insights and more in-depth discussions have been added to the main text and its supporting information.

I recommend its publication.

Reviewer #2 (Remarks to the Author):

Well done. The authors responded to the reviewers questions, suggestions, and concerns appropriately. Congratulations to this work well done.

Reviewer #1 (Remarks to the Author):

Thank you for considering all the comments and suggestions provided. The manuscript has been revised accordingly and new insights and more in-depth discussions have been added to the main text and its supporting information.

I recommend its publication.

Response: We thank the reviewer for the kind comment and recommendation.

Reviewer #2 (Remarks to the Author):

Well done. The authors responded to the reviewers questions, suggestions, and concerns appropriately. Congratulations to this work well done.

Response: We are thankful for the reviewer's approval and the constructive comments on the manuscript. The feedback helped us improve the manuscript much.